# What influences home delivery among women who live in urban areas? Analysis of 2014 Ghana Demographic and Health Survey data

**Bright Opoku Ahinkorah**[1][☯], **Abdul-Aziz Seidu**[2][☯], **Eugene Budu**[2]*, **Ebenezer Agbaglo**[3][‡],
**Francis Appiah**[2][‡], **Collins Adu**[4][‡], **Anita Gracious Archer**[5][‡], **Edward Kwabena Ameyaw**[1][☯]

1 The Australian Centre for Public and Population Health Research (ACPPHR), Faculty of Health, University of Technology Sydney, Sydney, Australia, 2 Department of Population and Health, University of Cape Coast, Cape Coast, Ghana, 3 Department of English, University of Cape Coast, Cape Coast, Ghana, 4 Department of Health Promotion and Disability Study, Kwame Nkrumah University of Science and Technology, Kumasi, Ghana, 5 School of Nursing and Midwifery, University of Health and Allied Sciences, Ho, Ghana

☯ These authors contributed equally to this work.
‡ EA, FA, CA and AGA also contributed equally to this work.
* budueugene@gmail.com

**Data Availability Statement:** Data is available from https://dhsprogram.com/.

## Abstract

### Background

In Ghana, home delivery among women in urban areas is relatively low compared to rural areas. However, the few women who deliver at home in urban areas still face enormous risk of infections and death, just like those in rural areas. The present study investigated the factors associated with home delivery among women who live in urban areas in Ghana.

### Materials and methods

Data for this study was obtained from the 2014 Ghana Demographic and Health Survey. We used data of 1,441 women who gave birth in the 5 years preceding the survey and were dwelling in urban areas. By the use of Stata version 14.2, we conducted both descriptive and multivariable logistic regression analyses.

### Results

We found that 7.9% of women in urban areas in Ghana delivered at home. The study revealed that, compared to women who lived in the Northern region, women who lived in the Brong Ahafo region [AOR = 0.38, CI = 0.17–0.84] were less likely to deliver at home. The likelihood of home delivery was high among women in the poorest wealth quintile [AOR = 2.02, CI = 1.06–3.86], women who professed other religions [AOR = 3.45; CI = 1.53–7.81], and those who had no antenatal care visits [AOR = 7.17; 1.64–31.3]. Conversely, the likelihood of home delivery was lower among women who had attained secondary/higher education [AOR = 0.30; 0.17–0.53], compared to those with no formal education.

**Funding:** The author(s) received no specific funding for this work.

**Competing interests:** The authors have declared that no competing interests exist.

**Abbreviations:** ANC, Antenatal Care; AOR, Adjusted Odds Ratio; CI, Confidence Interval; DHS, Demographic and Health Surveys; MMR, Maternal Mortality Ratio; WHO, World Health Organization; SDG, Sustainable Development Goal; SSA, sub-Saharan Africa; LMICs, Low and Middle-income countries; PNG, Papua New Guinea.

## Conclusion

The study identified region of residence, wealth quintile, religion, antenatal care visits, and level of education as factors associated with home delivery among urban residents in Ghana. Therefore, health promotion programs targeted at home delivery need to focus on these factors. We also recommend that a qualitative study should be conducted to investigate the factors responsible for the differences in home delivery in terms of region, as the present study could not do so.

## Background

Reducing maternal mortality has been one of the greatest public health concerns. In line with this, the United Nations instituted the Millennium Development Goals (MDGs) in 2000 with Goal 5 aimed at reducing maternal mortality by 75% by the year 2015 [1, 2]. Although there was a significant decline in maternal mortality rates in countries all over the world, Ghana could not achieve this target, as maternal mortality reduced only by 45% by the end of 2015 [3]. In 2015, the United Nations came up with the Sustainable Development Goals (SDGs), which also targets improvements in women's health, including reduction of maternal mortality. Specifically, target 3.1 of the SDGs aims at reducing the global maternal mortality ratio to less than 70 per 100,000 live births by the year 2030 [4, 5]. Despite this, the world still struggles with maternal mortality, with sub-Saharan African countries disproportionately affected [5], with 351 deaths per 100,000 live births [6]. In the context of Ghana, as at 2017, the maternal mortality ratio stood at 310 deaths per 100,000 births [7].

In low- and middle-income countries (LMICs), several significant efforts have been made to reduce maternal deaths through enhanced maternal healthcare services utilization globally [8–10]. However, many childbearing women in LMICs, especially countries in sub-Saharan Africa (SSA), still face challenges in accessing and utilizing maternal healthcare services, including delivery services, and opt to deliver at home [11, 12]. Evidence suggests that home births among women in SSA pose high risks to the health of the mother and the child during the period after delivery [13–15]. Some of these risks include desertion of colostrum provision and breastfeeding practices, neglect of immunisations and nutrition supplementation for mother and child, and lack of postnatal care check-up for the child and mother [16–18].

A larger percentage of maternal mortalities in Ghana is caused by pregnancy-related issues such as obstetric complications, which result in death during pregnancy, childbirth, or within 42 days after delivery [5]. This implies that maternal mortality can be reduced by ensuring that women have better maternal healthcare services, including health facility delivery [5]. Despite the importance of health facility delivery, some pregnant women in the country still deliver at home. Home delivery has been defined as any birth that had taken place in a pregnant woman's home or the homes of other people [19, 20]. In Ghana, several studies have shown that pregnant women who deliver at home often receive unskilled assistance during delivery [20–24]. Unskilled assistance has been defined as delivery assistance provided by traditional birth attendants (TBAs), relatives, and friends instead of skilled birth attendants (SBAs) such as doctors, physician assistants, midwives, or nurses [20, 21, 25, 26].

Over the years, the government of Ghana has attempted to improve access to maternal healthcare services. In 2003, the government, for example, introduced the waiver of delivery fees, and by 2005, fees on delivery care were abolished in all the then 10 regions of the country [27]. This was followed by the introduction of the National Health Insurance Scheme (NHIS)

in 2005. The NHIS allows all pregnant women under the scheme to have free access to maternal healthcare services, including antenatal care, delivery services, postnatal care, and neonatal care [28]. This program saw a decrease in home delivery from 45% in 2007 to 20% in 2017 [7]. However, there are rural-urban differentials in terms of home delivery, with the prevalence of home delivery in rural areas being 40%, as compared to 10.2% in urban areas [29]. Despite the seemingly low prevalence of home delivery in urban areas, the few women who deliver at home in urban areas still face the same risk of infections and death, just like those in rural areas since most of the urban women who deliver at home are assisted by unskilled birth attendants such as TBAs, relatives, and friends [30]. Besides, with access to health facilities in urban areas and waiver of delivery fees [31, 32], it is expected that urban women in Ghana will not consider home delivery. What is, therefore, unclear is why urban women in the country will still deliver at home in the midst of the opportunities to access health facility delivery.

It is, therefore, important to understand the factors associated with home delivery among urban women in Ghana and provide useful information for interventions aimed at reducing maternal mortality in the country. In Ghana, a few studies have been conducted in this regard. Studies by Ganle et al. [24] and Boah et al. [23] in Northern Ghana and the Builsa South District of the Upper West region, respectively, identified fewer antenatal care visits, lack of health insurance, living in a male-headed household, being unexposed to media, parity, poor attitude of nurses, lack of transportation, cost of delivery kits, traditional beliefs and practices as predictors of home delivery. A recent study also identified women's region of residence, educational status, wealth quintile, parity, religion, number of antenatal care (ANC) visits, and NHIS coverage as predictors of women's use of home birth services [20]. However, these studies focused on women in rural areas [20, 23, 24], ignoring the fact that a considerable number of women in urban areas still deliver at home and receive unskilled assistance during delivery. The present study departs from the previous ones by assessing the factors associated with home delivery among women in urban areas in Ghana. Findings from the study will help to enhance advocacy and educational strategies like, peer teaching, and mentor-mentee programmes at both national and community levels for women to encourage health facility delivery. Such interventions will help the country to contribute significantly to achieving target 1 of the Sustainable Development Goal 3 that aims to reduce global maternal mortality ratio to less than 70 per 100,000 live births by 2030.

## Materials and methods

### Study setting

The study setting is the Republic of Ghana. The Republic of Ghana is one of the countries in West Africa and has a total land area of 238,533 square kilometres [33]. It is bounded by Burkina Faso on the north, Togo in the east and Côte d'Ivoire on the west. According to the National Population Census conducted in 1960, 1970, 1984, 2000, and 2010, the population of the country stood at 6,726,815; 8,559,313; 12,296,081; 18,912,079, and 24,658,823 respectively [33]. As at the time of the surveys, the country was divided into ten regions, namely, Western, Central, Greater Accra, Volta, Eastern, Ashanti, Brong Ahafo, Northern, Upper East, and Upper West. However, currently, the country has 16 regions. The 16 regions are Oti, Brong Ahafo, Bono East, Ahafo, North East, Savannah, Western North, Western, Greater Accra, Central, Eastern, Upper East, Upper West, Volta, Northern, and Ashanti. The country is urbanized with about 51 percent of its population living in urban areas and 49 percent in rural areas. In terms of ethnicity, the dominant ethnic groups in the country are Akan (47.5%), Mole Dagbani (16.6%), Ga-Adangbe (7.4%), Gruma (5.7%), Guan (3.7%), Grusi (2.5%), with the rest belonging to 'other' ethnic groups namely Mande, Hausa and other ethnic groups. In terms of

religion, the majority of Ghanaians (71.2%) are Christians (Catholic, Protestant, Pentecostal/ Charismatic, and other Christian), followed by Islam (17.6%), Traditionalist (5.2%), No Religion (5.3%), and 'Other' religion (0.8%) [33].

## Data source

The study used data from the 2014 Ghana Demographic and Health Survey (GDHS). DHS is a nationwide survey collected every five-year period across low- and middle-income countries. In this study, the children's file was used and it contains the responses of children under five years born to women aged 15 and 49. The survey targets core maternal and child health indicators such as place of delivery.

## Study design

The 2014 GDHS employed a cross-sectional study design in gathering data from the respondents. There are four Model Questionnaires in the GDHS: a Household Questionnaire, a Woman's Questionnaire, a Man's Questionnaire, and a Biomarker Questionnaire. Each of these questionnaires gather data from households, men, and women. In Ghana, the DHS has been conducted in 1988, 1993, 1998, 2003, 2008, and 2014.

## Sampling procedure/sample size determination

Stratified dual-stage sampling approach was employed in the survey. The first step involved the selection of clusters across urban and rural locations from the entire nation. These made up enumeration areas (EAs) for the study. These clusters were selected from the erstwhile 10 administrative regions of the country and across urban (n = 216) and rural (n = 211) areas. This was followed by a systematic household sampling within the selected EAs. This constituted a total sample size of 12,831 households. From the 12,831 households, a total of 9,396 women (response rate, 97.3%) were interviewed for the survey. For the purpose of this study, only women in the urban areas who had information on birth history in the 5 years preceding the survey and complete cases on all the variables considered for the study were included (N = 1,441). Details of the methodology employed by the GDHS can be found in the final report [16].

## Data collection/quality assurance

The 2014 GDHS gathered data from men and women. Data collection was done by survey staff who are trainees and are given instructions in standard DHS procedures. These procedures include general interviewing techniques, conducting interviews at the household level, measuring blood pressures, review of each question, and mock interviews between participants. To ensure participants understood the questions being asked, the definitive questionnaires were first prepared in English and subsequently translated by experts into the major local languages at the various data collection points. Interviews are also conducted in local languages. As part of quality assurance, ten women and five men participated in a pretest training and field practice of the GDHS protocol and instruments over a three-week period, 9–28 June, 2014. The pretest participants were later used as field supervisors or editors, or as field coordinators to facilitate the data collection during the main fieldwork. Field staff were further given training before the actual data collection to ensure that they are able to gain accurate understanding of the data collection instruments [34].

## Variables studied

**Outcome variable.** The outcome variable considered in this study was "home delivery". This variable was obtained from the question, "Where did you deliver [name]?" In the GDHS, responses to this question were home, other home, government hospital, government health centre/clinic, government health post/ Community-based Health Planning and Services (CHPS), other public, private hospital/clinic, maternity homes, and others. These responses were dichotomised into health facility delivery = 0 and home delivery = 1. Home delivery referred to deliveries that occurred in respondent's home and other home. On the other hand, deliveries that occurred at government hospital, government health centre/clinic, government health post/CHPS, other public, private hospital/clinic, maternity homes, and other health facilities were grouped as "health facility delivery."

**Explanatory variables.** The study considered twelve explanatory variables. These are age, region, religion, ethnicity, educational level, marital status, wealth status, employment, parity, sex of household head, ANC visits, and decision-making for healthcare. These variables were not determined a priori; instead, they were determined based on parsimony, theoretical relevance, and practical significance with place of delivery [35–37]. The categorization of the variables can be found in Table 1.

**Data processing procedures and analyses.** The statistical software Stata version 14.0 was used to process the data. Both bivariate and multivariate analyses were employed in this study and results were tested at 95% confidence interval. Bivariate analysis was conducted to show the proportion of home deliveries across socio-demographic characteristics with their significance levels and chi-square values ($\chi2$). Multivariable analysis (binary logistic regression) was further conducted. Only the variables that showed statistical significance in the bivariate analysis were included in the regression analysis. Before the binary logistic regression analysis, we conducted a multicollinearity test of all the statistically significant variables using the variance inflation factor (VIF), and it showed no evidence of collinearity among the explanatory variables (Mean VIF = 1.32, Max VIF = 1.57, Minimum = 1.02). The results were presented as adjusted odds ratios, with their corresponding 95% confidence intervals signifying their level of precision. Statistical significance was declared at $p<0.05$. Sample weight was applied and the survey command (svy) was used to account for the complex sampling design of the survey. We wrote the manuscript following the "Strengthening the Reporting of Observational Studies in Epidemiology" (STROBE) statement.

## Ethical approval

The survey reported that ethical approval was granted by the Institutional Review Board of ICF International and Ethical Review Committee of Ghana Health Service [16]. We further obtained permission from the DHS Program for use of this data for the study. The data is available on https://dhsprogram.com/data/dataset/Ghana_Standard-DHS_2014.cfm?flag=0

## Results

### Distribution of the prevalence of home deliveries among women in the urban areas

Table 1 presents results on the distribution of the prevalence of home deliveries among women in the urban areas of Ghana across socio-demographic characteristics. The prevalence of home deliveries in the country was 7.9%, with variations across the various socio-demographic characteristics of the respondents. The results of the chi-square test showed that

**Table 1.** Weighted distribution of the prevalence of home deliveries among women in the urban areas of Ghana across socio-demographic characteristics (n = 1,441).

| Variable | Frequency (n) | Percentage (%) | Home delivery (n, %) | Health facility delivery (n, %) | $\chi^2$ (p-value) |
|---|---|---|---|---|---|
| **Age** | | | | | 0.57 (0.75) |
| 15–24 | 174 | 12.1 | 17 (10.0) | 157 (90.0) | |
| 25–34 | 787 | 55.6 | 61 (7.7) | 726 (92.3) | |
| 35 years or more | 481 | 33.4 | 36 (7.5) | 444 (92.5) | |
| **Region** | | | | | 101.64 (p<0.001) |
| Western | 109 | 7.5 | 68 (6.3) | 102 (93.7) | |
| Central | 107 | 7.4 | 11 (9.8) | 97 (90.2) | |
| Greater Accra | 437 | 30.4 | 16 (3.7) | 421 (96.3) | |
| Volta | 80 | 5.6 | 13 (16.1) | 67 (83.9) | |
| Eastern | 118 | 8.2 | 10 (8.5) | 108 (91.6) | |
| Ashanti | 333 | 23.1 | 15 (4.4) | 319 (95.6) | |
| Brong Ahafo | 116 | 8.1 | 8 (7.0) | 108 (93.0) | |
| Northern | 94 | 6.5 | 32 (34.3) | 62 (65.7) | |
| Upper East | 31 | 2.1 | 3 (7.4) | 28 (92.6) | |
| Upper West | 15 | 1.0 | 1 (1.5) | 14 (98.5) | |
| **Occupation** | | | | | 0.04 (0.840) |
| Working | 238 | 16.5 | 95 (7.9) | 1107 (92.1) | |
| Not working | 1203 | 83.5 | 19 (8.0) | 219 (92.0) | |
| **Ethnicity** | | | | | 16.32 (p<0.001) |
| Akan | 748 | 51.9 | 39 (5.3) | 709 (94.7) | |
| Ga/Dangme | 131 | 9.1 | 10 (7.7) | 121 (92.4) | |
| Mole Dagbani | 235 | 16.1 | 34 (14.4) | 199 (85.6 | |
| Other | 359 | 22.8 | 31 (9.5) | 298 (90.5) | |
| **Educational level** | | | | | 130.98 (p<0.001) |
| No education | 222 | 15.4 | 51 (23.1) | 170 (76.9) | |
| Primary | 212 | 14.7 | 32 (14.9) | 180 (85.1) | |
| Secondary/Higher | 1,007 | 69.9 | 31 (3.1) | 976 (96.9) | |
| **Wealth index** | | | | | 153.31 (p<0.001) |
| Poorest | 67 | 4.7 | 23 (33.5) | 45 (66.5) | |
| Poorer | 80 | 5.5 | 26 (32.5) | 54 (67.5) | |
| Middle | 232 | 16.1 | 31 (13.3) | 201 (86.7) | |
| Richer | 443 | 30.7 | 23 (5.2) | 420 (94.8) | |
| Richest | 619 | 43.0 | 12 (1.9) | 607 (98.1) | |
| **Parity** | | | | | 30.30 (p<0.001) |
| One birth | 292 | 20.3 | 7 (2.5) | 285 (97.5) | |
| Two births | 364 | 25.2 | 18 (5.1) | 345 (94.9) | |
| Three births | 299 | 20.8 | 29 (9.6) | 271 (90.4) | |
| Four or more births | 486 | 33.7 | 60 (12.3) | 426 (87.7) | |
| **Religion** | | | | | 88.46 (p<0.001) |
| Christianity | 1151 | 79.9 | 62 (5.4) | 1090 (94.6) | |
| Islam | 247 | 17.1 | 36 (14.7) | 210 (85.3) | |
| Other | 43 | 3.0 | 16 (38.0) | 27 (62.0) | |
| **Marital status** | | | | | 0.72 (0.397) |
| Married | 1084 | 75.2 | 83 (7.7) | 1000 (92.3) | |
| Cohabiting | 357 | 24.8 | 31 (8.6) | 327 (91.4) | |
| **Sex of household head** | | | | | 0.68 (0.411) |

*(Continued)*

**Table 1.** (Continued)

| Variable | Frequency (n) | Percentage (%) | Home delivery (n, %) | Health facility delivery (n, %) | $\chi^2$ (p-value) |
|---|---|---|---|---|---|
| Male | 1141 | 79.1 | 89 (7.8) | 1051 (92.2) | |
| Female | 300 | 20.9 | 25 (8.3) | 275 (91.7) | |
| **ANC visits** | | | | | 50.48 (p<0.001) |
| No ANC visits | 16 | 1.1 | 8 (49.2) | 8 (50.8) | |
| Had ANC visit | 1425 | 98.9 | 106 (7.5) | 1319 (92.5) | |
| **Healthcare decision-making** | | | | | |
| Not alone | 1035 | 71.8 | 81 (7.9) | 953 (92.1) | |
| Respondent alone | 406 | 28.2 | 33 (8.1) | 373 (91.9) | |
| National (Total) | 1,441 | 100 | 114 (7.9) | 1327 (92.1) | |

Source: Computed from 2014 GDHS.

region, ethnicity, educational level, wealth index, parity, religion, and ANC visits had significant associations with home delivery (p<0.001).

## Factors associated with home delivery among women who live in the urban areas of Ghana

Table 2 shows results on the factors associated with home delivery among women in the urban areas of Ghana. Compared to women who lived in the Northern region, women who lived in the Brong Ahafo region [AOR = 0.38, CI = 0.17–0.84] were less likely to deliver at home. Women with poorest wealth quintile were more likely to deliver at home, compared to those with middle wealth quintile [AOR = 2.02, CI = 1.06–3.86]. The likelihood of home delivery was higher among women who professed other religions, compared to Christians [AOR = 3.45; CI = 1.53–7.81]. Home delivery was found to be higher among women who had no ANC visits, compared to those who had at least one ANC visit [AOR = 7.17; CI = 1.64–31.3]. Conversely, the likelihood of home delivery was lower among women who had attained secondary/higher education [AOR = 0.30; CI = 0.17–0.53], compared to those with no formal education.

## Discussion

Home delivery presents an array of negative health complications to the mother and the child. Hence, ensuring health facility delivery has the potential to avert such risks [5]. The primary focus of our study was to assess the factors associated with home delivery in urban Ghana. We found that 7.9% of urban women in Ghana deliver at home. Region, wealth quintile, religion, ANC visits, and level of education were found as factors associated with home delivery among urban women in Ghana. Our study revealed that, compared to all the regions, women who lived in the Northern region were more likely to deliver at home. Debatably, health facilities in Ghana are not equally distributed across all the regions of the country. Typically, the northern part of the country is less endowed with health facilities [38, 39].

Several studies have further remarked that accessibility and availability of quality maternal health services influence the use of an institution for delivery [40–44]. Shahabuddin et al. [45] similarly noted that young women from mountain region Nepal were less likely to choose institutional delivery, compared with women in the Terai region. This implies that, without equitable distribution of health facilities and elimination of accessibility barriers including

**Table 2. Logistic regression analysis on predictors of home delivery among women in the urban areas of Ghana.**

| Variable | | 95% CI | |
|---|---|---|---|
| | AOR | Lower Bound | Upper Bound |
| **Region** | | | |
| Western | 0.63 | 0.20 | 2.01 |
| Central | 0.74 | 0.29 | 1.87 |
| Greater Accra | 0.42 | 0.14 | 1.28 |
| Volta | 0.80 | 0.34 | 1.90 |
| Eastern | 0.54 | 0.21 | 1.38 |
| Ashanti | 0.59 | 0.23 | 1.49 |
| Brong Ahafo | 0.38* | 0.17 | 0.84 |
| Northern | **Ref** | **Ref** | **Ref** |
| Upper East | 0.26** | 0.11 | 0.60 |
| Upper West | 0.15** | 0.04 | 0.62 |
| **Ethnicity** | | | |
| Akan | 1.39 | 0.69 | 2.79 |
| Ga/Dangme | 1.71 | 0.55 | 3.36 |
| Mole Dagbani | **Ref** | **Ref** | **Ref** |
| Other | 0.86 | 0.48 | 1.56 |
| **Education** | | | |
| No education | **Ref** | **Ref** | **Ref** |
| Primary | 0.80 | 0.46 | 1.40 |
| Secondary/Higher | 0.30*** | 0.17 | 0.53 |
| **Wealth** | | | |
| Poorest | 2.02* | 1.06 | 3.86 |
| Poorer | 1.84 | 0.99 | 3.41 |
| Middle | **Ref** | **Ref** | **Ref** |
| Richer | 0.38*** | 0.21 | 0.68 |
| Richest | 0.18*** | 0.08 | 0.42 |
| **Parity** | | | |
| One birth | 0.52 | 0.26 | 1.02 |
| Two births | 0.96 | 0.54 | 1.69 |
| Three births | 1.36 | 0.80 | 2.31 |
| Four or more births | **Ref** | **Ref** | **Ref** |
| **Religion** | | | |
| Christianity | **Ref** | **Ref** | **Ref** |
| Islam | 0.97 | 0.53 | 1.75 |
| Other | 3.45*** | 1.53 | 7.81 |
| **ANC visits** | | | |
| No ANC visits | 7.17** | 1.64 | 31.3 |
| Had ANC visit | **Ref** | **Ref** | **Ref** |
| N | 1441 | | |
| Pseudo $R^2$ | 0.259 | | |

Exponentiated coefficients; 95% confidence intervals in brackets

*$p < 0.05$

** $p < 0.01$

*** $p < 0.001$, aOR = adjusted odds ratios.

provision of efficient and effective referral services, health facility delivery will be cumbersome for most women in Ghana who are in the disadvantaged regions.

We found that women with poorest wealth quintile had higher odds to deliver at home, compared to those with middle wealth quintile. Our results are in line with previous studies in other LMICs such as Nepal [27, 45], Malawi [46], and Guinea-Bissau [47]. It is increasingly known that wealthier women are more likely to deliver in healthcare facilities than their poorer counterparts [48–50]. What might have caused this disparity in institutional delivery among the rich and the poor could possibly be attributed to financial stands. Poor women might be challenged financially when there is the need to deliver in a health facility, including the cost of transport and buying other items needed for delivery.

Our study also revealed that those who professed other religions had higher propensity to home delivery, compared to Christians. Religious affiliation was responsible for the disparity in institutional delivery in other studies. For instance, in Nepal, Shahabuddin et al. [45] observed that young Muslim women were 1.82 times more likely to deliver at an institution, compared with young Hindu women. We must acknowledge that religious affiliation comes with specific beliefs and practices which may influence women's general practices including opting for health facility delivery [23, 24]. Our study fails to unravel why women from other religious backgrounds were more likely to deliver at home, compared to Christians. We, therefore, suggest that a qualitative study be conducted on religion and place of delivery to understand the phenomenon better.

It is common knowledge that ANC utilisation has a spilt over effect on mothers' choice of place of delivery, whereby women who utilize ANC mostly prefer institutional delivery supervised by health professionals [51, 52]. As such, it was not surprising that home delivery was found to be higher among women who had no ANC visits, as compared to those who had at least one ANC visit in our current study. A systematic review in assessing factors influencing utilisation of maternal health services by adolescent mothers in LMICs concluded that the use of ANC predicted use of skilled birth attendance, and postnatal care [53]. Practically, it is suggestive that sufficient ANC utilisation is likely to increase pregnant women's awareness of possible complications and safe delivery practices, which consequently urges them to deliver in a health facility [54–57]. It has also been argued that women who visit health facilities for healthcare services including ANC check-ups might be exposed to counselling and guidance from health professionals [58]. Both scenarios enlighten them about the dangers associated with home deliveries. Theoretically, following the proposition raised by the protection motivation theory, women, having been exposed to vital information during ANC, will opt for delivering at a health facility to serve as a protection against unforeseen predicaments associated with home delivery [59–61].

Finally, we noted that the probability to deliver at home was low among women who had attained secondary/higher level of education, compared to those with no formal education. Similarly, studies in Malawi observed that women who had no formal education were about four-fold probable to deliver at home, compared to women who had attained secondary or higher level [46]. In a related study, Yaya, Bishwajit, and Gunawardena [47] also found that, among residents in urban areas of Guinea-Bissau, compared those who had no education, those who had primary and secondary/higher level of education were 2.2 and 3.3 times more likely to deliver at a health facility. Level of education has been a determinant to home delivery in Zambia as well, and it was known that women who had four years of schooling or less were 63 percent more likely to deliver at home than a health facility, compared to those who had at least five years of schooling. A recent study also found that having education increases the likelihood that a woman would decide to deliver her baby in a hospital or maternity home than at home or in other places [62]. Oyedele [62] further explained that education increases

individuals' awareness about health holistically and exposes them to benefits associated with complication prevention. Therefore, we can argue that the educated Ghanaians would be compelled to live a sanitary and hygienic life including opting institutional delivery.

## Strengths and limitations

Among the strengths of the study is the fact that it depended on a nationally representative data set. Also, the study adopted probability method in selecting survey respondents and used sound analytical procedure which enhances the robustness of the results. However, our results should be interpreted with caution since causality cannot be established.

## Conclusion

The major factors associated with home delivery among urban residents observed in our study were region, wealth quintile, religion of affiliation, ANC visits, and educational attainment. What might have compelled Northern residents to be inclined to home delivery calls for health education emphasising health facility delivery while prioritising health facility delivery on policy initiatives.

The significance of enhancing health facility delivery and reducing if not eliminating home delivery among childbearing women cannot be ignored if Ghana can contribute in achieving SDG 3.1 which aims to reduce global maternal mortality ratio to less than 70 per 100,000 live births by 2030. The findings call for the need to empower women at both national and community levels to utilise ANC, using sufficient counselling, guidance, and mass sensitisation through various mass media platforms. It is important for future studies to employ qualitative design to provide a deeper understanding of some of the findings in the current study.

## Acknowledgments

We acknowledge Measure DHS for providing us with the data.

## Author Contributions

**Conceptualization:** Bright Opoku Ahinkorah, Abdul-Aziz Seidu, Edward Kwabena Ameyaw.

**Data curation:** Bright Opoku Ahinkorah, Abdul-Aziz Seidu, Eugene Budu, Edward Kwabena Ameyaw.

**Formal analysis:** Bright Opoku Ahinkorah, Abdul-Aziz Seidu, Eugene Budu, Edward Kwabena Ameyaw.

**Funding acquisition:** Bright Opoku Ahinkorah, Abdul-Aziz Seidu, Edward Kwabena Ameyaw.

**Investigation:** Bright Opoku Ahinkorah, Abdul-Aziz Seidu, Edward Kwabena Ameyaw.

**Methodology:** Bright Opoku Ahinkorah, Abdul-Aziz Seidu, Edward Kwabena Ameyaw.

**Project administration:** Bright Opoku Ahinkorah, Abdul-Aziz Seidu, Edward Kwabena Ameyaw.

**Resources:** Bright Opoku Ahinkorah, Abdul-Aziz Seidu, Edward Kwabena Ameyaw.

**Software:** Bright Opoku Ahinkorah, Abdul-Aziz Seidu, Edward Kwabena Ameyaw.

**Supervision:** Bright Opoku Ahinkorah, Abdul-Aziz Seidu, Edward Kwabena Ameyaw.

**Validation:** Bright Opoku Ahinkorah, Abdul-Aziz Seidu, Edward Kwabena Ameyaw.

**Visualization:** Bright Opoku Ahinkorah, Abdul-Aziz Seidu, Edward Kwabena Ameyaw.

**Writing – original draft:** Bright Opoku Ahinkorah, Abdul-Aziz Seidu, Ebenezer Agbaglo, Francis Appiah, Collins Adu, Anita Gracious Archer, Edward Kwabena Ameyaw.

**Writing – review & editing:** Bright Opoku Ahinkorah, Abdul-Aziz Seidu, Ebenezer Agbaglo, Francis Appiah, Collins Adu, Anita Gracious Archer, Edward Kwabena Ameyaw.

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
