## [Decision Letter · Decision Letter 0]

13 Nov 2020

PONE-D-20-23864

What influences home delivery among women who live in urban areas in Ghana? Analysis of 2014 Ghana Demographic and Health Survey data

PLOS ONE

Dear Dr. Budu,

Thank you for submitting your manuscript to PLOS ONE. After careful consideration, we feel that it has merit but does not fully meet PLOS ONE’s publication criteria as it currently stands. Therefore, we invite you to submit a revised version of the manuscript that addresses the points raised during the review process.

Three experts in the field handled your manuscript, and we are very thankful for their time and efforts. Although interest was found in your study, some comments arose the need addressing in your revised manuscript.

We look forward to receiving your revised manuscript.

Kind regards,

Frank T. Spradley

Academic Editor

PLOS ONE

3. We noticed you have some minor occurrence of overlapping text with the following previous publication, which needs to be addressed:

- https://journals.plos.org/plosone/article/file?id=10.1371%2Fjournal.pone.0220970&type=printable

In your revision ensure you cite all your sources (including your own works), and quote or rephrase any duplicated text outside the methods section. Further consideration is dependent on these concerns being addressed.

Reviewers' comments:

Reviewer's Responses to Questions

**Comments to the Author**

1. Is the manuscript technically sound, and do the data support the conclusions?

Reviewer #1: Yes

Reviewer #2: Yes

Reviewer #3: Yes

2. Has the statistical analysis been performed appropriately and rigorously? 

Reviewer #1: Yes

Reviewer #2: Yes

Reviewer #3: Yes

3. Have the authors made all data underlying the findings in their manuscript fully available?

Reviewer #1: Yes

Reviewer #2: Yes

Reviewer #3: Yes

4. Is the manuscript presented in an intelligible fashion and written in standard English?

Reviewer #1: Yes

Reviewer #2: No

Reviewer #3: Yes

5. Review Comments to the Author

Reviewer #1: Background information, the following are missing

1. Clear definition of home birth is missing

2. What is situation in Ghana, does home birth = unskilled assistance

3. How does this study define unskilled birth assistance?

4. It must come clearly why urban? The expectation is urban home birth may invite skilled attendant as compared to rural? 5. Are there traditional birth attendants in urban?

The paper is about predictors of home birth but authors have to say something on the proportion of home birth vs health facility birth.

Reviewer #2: Review report ( Dr. Habtamu Tolera)

The manuscript reports the findings regarding “What influences home delivery among women who live in urban areas in Ghana? Analysis of 2014 Ghana Demographic and Health Survey data (PONE-D-20-23864). This kind of study is much relevant in context of LMICs like Ghana. The sample size is adequate and the analyses is really very rigorous and well done. I have found that authors strongly need to rework on the background and methodology sections. They also need to avoid long sentences used across the manuscript. Here below in my report I have tried to review some points or concerns that need authors to further rework on them to improve the power of this manuscript to get published in PLoS One Journal. Authors can also refer to the attached pdf file.

Affiliation:

Corresponding author's E-mail address is enough. So, remove others’ email addresses from the front page of the manuscript

Background:

• I have found the sentences in the first paragraph of the “Background” section of this study (page # 2, Lines 73-84) were too long and lacks clarity. Thus, I advise authors: (1) avoiding long sentences from this section and elsewhere across the manuscript, (2) clarify all so that they convey the correct information for potential readers of the manuscript.

• See Page # 3, line 97. If this is so, why do the author(s) interested to urban Ghana than rural Ghana in this study? authors need to give justifications for selection criteria.

• See also sentences on page 3, lines 103 through 110. They were not clearly stated for potential international readers, all have been needed to be rewritten.

• Page 3, Lines 110-112 said like this, “However, these studies focused more on rural areas, ignoring the fact that a considerable number of women in urban areas still use home delivery service” This needs authors to acknowledge (cite) supporting evidences to say so. So that they may convince the readers.

• Finally, your “Background” section needs more detail. I suggest that authors improve the description at "Background" section to provide more justification for their study (specifically, you should expand upon the knowledge gap being filled). I also recommend authors to read and review adequate empirical works on home delivery elsewhere in LMICs to enrich also their "Background" section. Why did you stick to works done in Ghana alone? They need to read vast works beyond urban Ghana and describe the contexts, background pictures (international or local pictures of the problem) of the topic; empirical/theoretical/methodological gaps observed with regard to the issue/the problem under study as well. Authors again need to state the benefits of this research findings (for program/policy interventions) in the last sentence of the "background" section or somewhere in this section.

Methods and materials

• I understand that authors analyzed GDHS or survey data collected by government. However, I have found that important methodological subsections/elements or components were missing in this study. So, authors need to add "Study setting", "study design", "Sampling procedure/sample size determination", "Data collection/quality assurance", "Data processing procedures and analyses". They need to customize these methodologies from GDHS "survey they used for this study. These are mandatory to be included under the methodological sections to be published in PLoS ONE.

• On Page 3, lines 119 through 121 Under “Data Source” section, it was stated/listed like this, “The survey targets core maternal and child health indicators such as unintended pregnancy, contraceptive use, skilled birth attendance, immunization among under-fives, and intimate partner violence”. Did these variables were captured/considered in the report/ analyses of this study? I have not they found them. If not relevant, please remove it If were not well addressed or if they were inserted by mistake.

• Line 133, better if you label it like this, "Variables studied" or “Measurements”

• Sentences from Lines 135 through 140, is not clear. The current phrasing makes comprehension difficult. so, have been rewritten.

• A sentence from Lines 135-38, were put like this, “The outcome variable employed for this study was “home delivery” which was obtained from the 136 question, “Where did you deliver?” Responses to this question were coded respondent’s home, other home, government hospital, government health centre/clinic, government health post/CHPS, other public, private hospital/clinic, maternity homes, and others”. My concern is that, under which category did authors assign or grouping "maternity home” in this coding of categories of the outcome variable, delivery choice status of study subjects. They need to recheck the coding

Results:

• Authors need to insert a new column in Tables 1 and 2 to report the frequency and the proportion of both home delivery and Non-home delivery statuses of women. Readers need to get these two values together AOR, CI, P-values, etc.

• Finally, some texts and paragraphs need editorial problems and authors need to rewrite with clear and plain language

Reviewer #3: This is an informative piece of work.

However, it would do better with inclusion of recent studies on home delivery in the introduction and or discussion to make it stronger.

Summarize all the key findings at the beginning of the discussion to get the reader a better perspective vs the discussion of results separately in fragmented style

The conclusion can still be strengthened, made more clearer and indicative.

Take care of references to ensure consistency, providing key links for critical documents and when these were accessed.

6. PLOS authors have the option to publish the peer review history of their article (what does this mean?). If published, this will include your full peer review and any attached files.

Reviewer #1: No

Reviewer #2: **Yes: **Habtamu Tolera

Reviewer #3: No

---

## [Author Response · Author response to Decision Letter 0]

16 Nov 2020

AUTHOR’S RESPONSE TO REVIEWS

Title: What influences home delivery among women who live in urban areas in Ghana? Analysis of 2014 Ghana Demographic and Health Survey data

Date: 16/11/2020

AUTHOR’S RESPONSE TO REVIEWS

Dear Editor and Reviewers,

We are pleased to resubmit for publication the revised version of “What influences home delivery among women who live in urban areas in Ghana? Analysis of 2014 Ghana Demographic and Health Survey data”. Thank you for giving us the opportunity to revise and resubmit this manuscript. We appreciate the time and constructive feedback provided by the reviewers and the Editor. We have added an additional author Anita Gracious Archer based on her valuable inputs during the revision process. The manuscript has certainly benefited from these insightful suggestions. Overall, the revision process is very productive. We have made every attempt to fully address all the comments in the revised manuscript. period. Based on the comments, we have responded specifically to each suggestion below. 

RESPONSE TO REVIEWS 

Reviewer #1: Background information, the following are missing

1. Clear definition of home birth is missing

2. What is situation in Ghana, does home birth = unskilled assistance

3. How does this study define unskilled birth assistance?

4. It must come clearly why urban? The expectation is urban home birth may invite skilled attendant as compared to rural? 5. Are there traditional birth attendants in urban?

Response: The background has been revised to incorporate all these suggestions (Line 99-104).

The paper is about predictors of home birth but authors have to say something on the proportion of home birth vs health facility birth. 

Response: We have mentioned that “We found that 7.9% of women in urbans areas in Ghana delivered at home” (Line 55).

Reviewer #2: Review report ( Dr. Habtamu Tolera)

The manuscript reports the findings regarding “What influences home delivery among women who live in urban areas in Ghana? Analysis of 2014 Ghana Demographic and Health Survey data (PONE-D-20-23864). This kind of study is much relevant in context of LMICs like Ghana. The sample size is adequate and the analyses is really very rigorous and well done. I have found that authors strongly need to rework on the background and methodology sections. They also need to avoid long sentences used across the manuscript. Here below in my report I have tried to review some points or concerns that need authors to further rework on them to improve the power of this manuscript to get published in PLoS One Journal. Authors can also refer to the attached pdf file.

Response: Thank you for your useful comments. We have considered all of them in the revised manuscript. 

Affiliation:

Corresponding author's E-mail address is enough. So, remove others’ email addresses from the front page of the manuscript

Response: We have taken out the E-mail addresses of the co-authors and left only that of the corresponding author. 

Background:

• I have found the sentences in the first paragraph of the “Background” section of this study (page # 2, Lines 73-84) were too long and lacks clarity. Thus, I advise authors: (1) avoiding long sentences from this section and elsewhere across the manuscript, (2) clarify all so that they convey the correct information for potential readers of the manuscript.

Response: We have clarified the information provided in that paragraph (Line 73-84)

• See Page # 3, line 97. If this is so, why do the author(s) interested to urban Ghana than rural Ghana in this study? authors need to give justifications for selection criteria.

Response: We have revised the background and provided justification for conducting the study using urban women. A recent study and a number of previous studies have focused on rural Ghana despite urban women who deliver at home still going through the same risk of infections and deaths as their rural counterparts (Line 113-120).

• See also sentences on page 3, lines 103 through 110. They were not clearly stated for potential international readers, all have been needed to be rewritten.

Response: We have revised those sentences (Line 121-130).

• Page 3, Lines 110-112 said like this, “However, these studies focused more on rural areas, ignoring the fact that a considerable number of women in urban areas still use home delivery service” This needs authors to acknowledge (cite) supporting evidences to say so. So that they may convince the readers.

Response: We have cited sources for the information (Line 130)

• Finally, your “Background” section needs more detail. I suggest that authors improve the description at "Background" section to provide more justification for their study (specifically, you should expand upon the knowledge gap being filled). I also recommend authors to read and review adequate empirical works on home delivery elsewhere in LMICs to enrich also their "Background" section. Why did you stick to works done in Ghana alone? They need to read vast works beyond urban Ghana and describe the contexts, background pictures (international or local pictures of the problem) of the topic; empirical/theoretical/methodological gaps observed with regard to the issue/the problem under study as well. Authors again need to state the benefits of this research findings (for program/policy interventions) in the last sentence of the "background" section or somewhere in this section.

Response: We have revised the background section to incorporate all these useful suggestions (Line 85-89; 121-138)

Methods and materials

• I understand that authors analyzed GDHS or survey data collected by government. However, I have found that important methodological subsections/elements or components were missing in this study. So, authors need to add "Study setting", "study design", "Sampling procedure/sample size determination", "Data collection/quality assurance", "Data processing procedures and analyses". They need to customize these methodologies from GDHS "survey they used for this study. These are mandatory to be included under the methodological sections to be published in PLoS ONE.

Response: We have considered these useful suggestions under “Materials and methods”.

• On Page 3, lines 119 through 121 Under “Data Source” section, it was stated/listed like this, “The survey targets core maternal and child health indicators such as unintended pregnancy, contraceptive use, skilled birth attendance, immunization among under-fives, and intimate partner violence”. Did these variables were captured/considered in the report/ analyses of this study? I have not they found them. If not relevant, please remove it If were not well addressed or if they were inserted by mistake.

Response: We have removed these from the paper (Line 160-163). 

• Line 133, better if you label it like this, "Variables studied" or “Measurements”

Response: We have now used variables studied (Line 194).

• Sentences from Lines 135 through 140, is not clear. The current phrasing makes comprehension difficult. so, have been rewritten.

Response: We have made the paragraph very clear (Line 196-204). 

• A sentence from Lines 135-38, were put like this, “The outcome variable employed for this study was “home delivery” which was obtained from the 136 question, “Where did you deliver?” Responses to this question were coded respondent’s home, other home, government hospital, government health centre/clinic, government health post/CHPS, other public, private hospital/clinic, maternity homes, and others”. My concern is that, under which category did authors assign or grouping "maternity home” in this coding of categories of the outcome variable, delivery choice status of study subjects. They need to recheck the coding

Response: In line with the definition of home delivery and health facility delivery in the DHS and in previous studies, ‘maternity home’ is considered part of health facility delivery in this study. 

Results:

• Authors need to insert a new column in Tables 1 and 2 to report the frequency and the proportion of both home delivery and Non-home delivery statuses of women. Readers need to get these two values together AOR, CI, P-values, etc.

Response: Per our understanding of your suggestion, we have inserted a column in Table 1 to report on the frequency and proportion of home delivery and health facility delivery. We are unsure if you also suggested we insert a column in Table 2 to report AOR, CI and P-values for health facility delivery as this is not appropriate since the outcome of interest in this study was “home delivery” and the AOR, CI and P-values produced in the regression analysis are only for the outcome of interest and not for both “home delivery” and “health facility delivery”. 

• Finally, some texts and paragraphs need editorial problems and authors need to rewrite with clear and plain language

Response: We have addressed all these issues. 

Reviewer #3: This is an informative piece of work.

However, it would do better with inclusion of recent studies on home delivery in the introduction and or discussion to make it stronger.

Summarize all the key findings at the beginning of the discussion to get the reader a better perspective vs the discussion of results separately in fragmented style

The conclusion can still be strengthened, made more clearer and indicative.

Take care of references to ensure consistency, providing key links for critical documents and when these were accessed.

Response: Thanks for your useful comments suggestions. All these issues have been considered in the revised paper (Line 73-138; 265-267; 335-345; 366-541)

---

## [Decision Letter · Decision Letter 1]

10 Dec 2020

PONE-D-20-23864R1

What influences home delivery among women who live in urban areas in Ghana? Analysis of 2014 Ghana Demographic and Health Survey data

PLOS ONE

Dear Dr. Budu,

Thank you for submitting your manuscript to PLOS ONE. After careful consideration, we feel that it has merit but does not fully meet PLOS ONE’s publication criteria as it currently stands. Therefore, we invite you to submit a revised version of the manuscript that addresses the points raised during the review process.

Carefully proof your manuscript to correct grammar or spelling errors.

We look forward to receiving your revised manuscript.

Kind regards,

Frank T. Spradley

Academic Editor

PLOS ONE

Reviewers' comments:

Reviewer's Responses to Questions

**Comments to the Author**

1. If the authors have adequately addressed your comments raised in a previous round of review and you feel that this manuscript is now acceptable for publication, you may indicate that here to bypass the “Comments to the Author” section, enter your conflict of interest statement in the “Confidential to Editor” section, and submit your "Accept" recommendation.

Reviewer #1: All comments have been addressed

Reviewer #2: All comments have been addressed

Reviewer #3: All comments have been addressed

2. Is the manuscript technically sound, and do the data support the conclusions?

Reviewer #1: Yes

Reviewer #2: Yes

Reviewer #3: Yes

3. Has the statistical analysis been performed appropriately and rigorously? 

Reviewer #1: Yes

Reviewer #2: Yes

Reviewer #3: Yes

4. Have the authors made all data underlying the findings in their manuscript fully available?

Reviewer #1: Yes

Reviewer #2: Yes

Reviewer #3: Yes

5. Is the manuscript presented in an intelligible fashion and written in standard English?

Reviewer #1: Yes

Reviewer #2: Yes

Reviewer #3: Yes

6. Review Comments to the Author

Reviewer #1: Authors have addressed the raised comments as required,

Minor revisions on the abstract and some typographical errors

Reviewer #2: I hope authors did all best. I appreciate your commitments. I have any concerns. I have finished in my side.

Reviewer #3: All concerns have been duly addressed. Information and narrative provided is sufficient. The data provided supports the conclusion provided.

7. PLOS authors have the option to publish the peer review history of their article (what does this mean?). If published, this will include your full peer review and any attached files.

Reviewer #1: No

Reviewer #2: **Yes: **Habtamu Tolera Deressa

Reviewer #3: No

---

## [Author Response · Author response to Decision Letter 1]

14 Dec 2020

AUTHORS’ RESPONSE TO EDITOR’S COMMENTS

Title: What influences home delivery among women who live in urban areas in Ghana? Analysis of 2014 Ghana Demographic and Health Survey data

Dear Editor,

Thank you for your email dated 10 December 2020 enclosing the Editor’s comments. We convey our gratitude to you for the comment that has led to the massive improvement of our paper entitled “What influences home delivery among women who live in urban areas in Ghana? Analysis of 2014 Ghana Demographic and Health Survey data”. We have now proofread the paper to correct grammar and spelling errors. All the changes are in tack changes in the revised manuscript. We believe the manuscript has improved substantively and will be published in your reputable journal. 

Version 2: PONE-D-20-23864R2

Date: 11/12/2020

Editor’s comment

Dear Dr. Budu,

Thank you for submitting your manuscript to PLOS ONE. After careful consideration, we feel that it has merit but does not fully meet PLOS ONE’s publication criteria as it currently stands. Therefore, we invite you to submit a revised version of the manuscript that addresses the points raised during the review process.

Carefully proof your manuscript to correct grammar or spelling errors.

Response: We have now proofread the paper to correct grammar and spelling errors 

Thank you

Sincerely

Eugene Budu

---

## [Editor Report · Decision Letter 2]

17 Dec 2020

What influences home delivery among women who live in urban areas in Ghana? Analysis of 2014 Ghana Demographic and Health Survey data

PONE-D-20-23864R2

Dear Dr. Budu,

We’re pleased to inform you that your manuscript has been judged scientifically suitable for publication and will be formally accepted for publication once it meets all outstanding technical requirements.

Kind regards,

Frank T. Spradley

Academic Editor

PLOS ONE

---

## [Editor Report · Acceptance letter]

23 Dec 2020

PONE-D-20-23864R2 

What influences home delivery among women who live in urban areas? Analysis of 2014 Ghana Demographic and Health Survey data 

Dear Dr. Budu:

I'm pleased to inform you that your manuscript has been deemed suitable for publication in PLOS ONE. Congratulations! Your manuscript is now with our production department. 

Kind regards, 

on behalf of

Dr. Frank T. Spradley 

Academic Editor

PLOS ONE